# Comparative Genomic Analysis of 450 Strains of *Salmonella*
*enterica* Isolated from Diseased Animals

**DOI:** 10.3390/genes11091025

**Published:** 2020-09-01

**Authors:** Shaohua Zhao, Cong Li, Chih-Hao Hsu, Gregory H. Tyson, Errol Strain, Heather Tate, Thu-Thuy Tran, Jason Abbott, Patrick F. McDermott

**Affiliations:** Center for Veterinary Medicine, U.S. Food and Drug Administration, Laurel, MD 20708, USA; cong.li@fda.hhs.gov (C.L.); Chih-Hao.Hsu@fda.hhs.gov (C.-H.H.); Gregory.Tyson@fda.hhs.gov (G.H.T.); Errol.Strain@fda.hhs.gov (E.S.); Heather.Tate@fda.hhs.gov (H.T.); Thu-Thuy.Tran@fda.hhs.gov (T.-T.T.); Jason.Abbott@fda.hhs.gov (J.A.); Patrick.McDermott@fda.hhs.gov (P.F.M.)

**Keywords:** *Salmonella*, pathogenicity islands, antimicrobial resistance, MDR plasmid

## Abstract

*Salmonella* is a leading cause of bacterial infections in animals and humans. We sequenced a collection of 450 *Salmonella* strains from diseased animals to better understand the genetic makeup of their virulence and resistance features. The presence of *Salmonella* pathogenicity islands (SPIs) varied by serotype. *S*. Enteritidis carried the most SPIs (*n* = 15), while *S*. Mbandaka, *S*. Cerro, *S.* Meleagridis, and *S*. Havana carried the least (*n* = 10). *S.* Typhimurium, *S.* Choleraesuis, *S*. I 4,5,12:i:-, and *S.* Enteritidis each contained the *spv* operon on IncFII or IncFII-IncFIB hybrid plasmids. Two *S.* IIIa carried a *spv* operon with *spvD* deletion on the chromosome. Twelve plasmid types including 24 hybrid plasmids were identified. IncA/C was frequently associated with *S.* Newport (83%) and *S*. Agona (100%) from bovine, whereas IncFII (100%), IncFIB (100%), and IncQ1 (94%) were seen in *S*. Choleraesuis from swine. IncX (100%) was detected in all *S*. Kentucky from chicken. A total of 60 antimicrobial resistance genes (ARGs), four disinfectant resistances genes (DRGs) and 33 heavy metal resistance genes (HMRGs) were identified. The *Salmonella* strains from sick animals contained various SPIs, resistance genes and plasmid types based on the serotype and source of the isolates. Such complicated genomic structures shed light on the strain characteristics contributing to the severity of disease and treatment failures in *Salmonella* infections, including those causing illnesses in animals.

## 1. Introduction

Currently, more than 2600 serotypes are recognized in *Salmonella*. Almost 99% of the serotypes associated with disease in warm-blooded animals and humans are members of *Salmonella enterica* subspecies *enterica* (subspecies I) [1]. They commonly elicit localized self-limiting inflammation of the terminal ileum and colon known as gastroenteritis in humans, and are also able to colonize and infect a broad range of animal species. Often, animal infections by this subspecies are asymptomatic; however, some serotypes can cause symptomatic disease, which is dependent on the infecting serotype and the species, genetic background, and immune status of the host. It is well known that the genetic background of pathogens plays a critical role in contributing to infection outcomes [2,3,4]. For example, *S.* Typhimurium can cause acute enteritis and septicemia in pigs and cattle yet colonizes the intestines of adult poultry asymptomatically [5].

*Salmonella* pathogenicity islands (SPIs) are well-studied genetic segments that contribute to infection outcomes. They contain many genes encoding virulence factors, including several types of secretion systems (T1SS, T3SS, and T6SS), adhesins, effector proteins, and other factors associated with bacterial invasion, enteropathogenesis, intracellular survival, and proliferation [6,7,8,9]. The acquisition of SPIs can be achieved through horizontal gene transfer from other bacterial species and is a “quantum leap” in *Salmonella* evolution [7,10,11]. Presently, 23 SPIs have been reported [4,7,12,13,14,15]. They are diverse in structure and function, and even the same SPI can vary within a certain serotype, suggesting a range of transmissible elements that are likely involved in the continuing evolution of SPIs and host specificity [9,10]. A study by Suez showed that among the first 21 SPIs, SPI-1 to -5, -9, -13 and -14 were conserved in invasive non-typhoidal *Salmonella* serovars (iNTS), while others were variable in different serotypes [9]. For example, *S*. Typhimurium LT2 carries 13 SPIs, whereas *S.* Typhi carries 17 SPIs [4]. Recently, we analyzed 61 *S*. Dublin strains isolated from human, sick bovine and retail beef along with 13 other serotypes, and the results showed that all the strains regardless of serotype carried SPI-1 to 6, 9 and 11. None of them carried SPI-7, 15, 20 and 21. All *S*. Dublin strains and one *S*. Enteritidis strain carried the most SPIs (*n* = 15) compared to other serotypes [16]. Previous studies of SPI profiles have been done by microarray, PCR, cloning, and Southern hybridizations or sequencing of ORFs of specific SPIs with limited numbers of isolates and serotypes. Such approaches have limitations and may provide an inaccurate picture of the structure and organization of SPIs. Since whole genome sequencing (WGS) technology has become faster and cheaper, it has become possible to comprehensively identify SPIs and other genomic features with large numbers of isolates and serotypes of interest.

In addition to SPIs, a virulence plasmid that carries a *spvRABCD* operon is also a significant virulence factor and contributes to the severity of the disease. Several studies have shown that *spv* genes can enhance the virulence of *Salmonella* by increasing the rate of bacterial replication within host cells and can help produce lethal disease in mice [17,18,19]. Libby et al. showed that *spv* genes are required for severe enteritis and systemic infection to occur in the natural host [3]. Virulence plasmids have been identified in host-adapted serotypes such as *S*. Dublin (bovine), *S.* Choleraesuis (swine), *S*. Paratyphi (human), *S*. Gallinarum and *S.* Pullorum (chicken), *S*. Abortusovis (sheep), and in some highly invasive strains of the broad-host serotypes, such as *S*. Typhimurium and *S*. Enteritidis [3,20,21].

*Salmonella* has become one of the most clinically important antimicrobial-resistant Gram-negative bacteria. Drug-resistant nontyphoidal *Salmonella* was listed as a serious threat to public health by the Centers for Disease Control and Prevention (CDC) (https://www.cdc.gov/drugresistance/biggest-threats.html). Antimicrobial resistance in *Salmonella* varies not only by serotype, but also by animal source and geographic location of isolates. Previously, we characterized antimicrobial susceptibility profiles of 380 *Salmonella* isolated from diseased animals with 16 antimicrobials [22]. Several serotypes such as *S*. Dublin, *S*. Choleraesuis, *S*. Agona, *S*. Uganda showed 100% resistance to ≥1 antimicrobial. The highly resistant (≥9 antimicrobials) serotypes were *S.* Uganda (100%), followed by *S*. Agona (79%), *S.* Newport (62%), *S.* Dublin (31%), and *S.* Anatum (25%). Regarding the source, 92% of isolates from swine showed resistance to ≥1 antimicrobial, followed by turkey (91%), bovine (77%) and chicken (68%) sources. However, bovine isolates showed the highest number of multi-drug resistance (MDR) and 35% of bovine isolates showed resistance to 9 or more antimicrobials tested [22].

The US National Antimicrobial Resistance Monitoring System (NARMS) has been monitoring antimicrobial resistance in *Salmonella* isolated from humans, food animals, and retail meats for the last 20 years. NARMS reports also show that resistance is strongly associated with the serotype and source of isolates. Based on the 2017 NARMS report, the most common MDR serotype in humans was *S*. I 4,[5],12:i:-, accounting for 25% of MDR *Salmonella* isolates. Approximately 80% of MDR *S.* I 4,[5],12:i- isolates showed resistance to ampicillin, streptomycin, sulfisoxazole, and tetracycline (ASSuT). For the food animals, there was a substantial increase in MDR *Salmonella* recovered from retail chickens (9.5 to 18%) and chicken cecal samples (15 to 25%) from 2015 to 2017 [23]. This increase was largely driven by the rise in MDR *S*. Infantis. We also noticed that the majority of MDR isolates in turkey were of serotypes *S*. Infantis, *S*. Reading, and *S*. I 4,[5],12:i:-. [23].

Increased antimicrobial resistance in *Salmonella* mainly results from acquisition of transmissible resistance genes on mobile elements such as plasmids, integrons, and transposons [24,25]. Based on a total of over 81,000 non-*Salmonella* genomes in the National Center for Biotechnology Information (NCBI) database, over 370 acquired antimicrobial resistance genes that encode resistance to β-lactam, aminoglycoside, fluoroquinolone, rifampicin, phenicol, trimethoprim, macrolides/lincosamides, polymyxin, sulfonamide, tetracycline, fosfomycin, and fusidic acid drug classes were identified [26]. Many previous studies have showed that plasmid types carrying specific genes were associated with *Salmonella* serotypes and isolate sources. For example, the MDR IncA/C plasmid that carried the *bla*_CMY-2_ gene was commonly seen in *S*. Dublin, *S*. Newport, and *S*. Typhimurium isolated from cattle, while *S*. Typhimurium isolated from chickens or chicken products with *bla*_CMY-2_ plasmids primarily had IncI1 plasmids [27]. Understanding the association between plasmid type and resistance genes, serotypes, and the source of the isolates is critical knowledge to support foodborne disease outbreak investigation, foodborne disease source attribution, as well as to understand antimicrobial resistance selection pressure from different food animal production environments. The objective of this study is to conduct comparative genomic analysis of *Salmonella* isolated from diseased animals and to elucidate the genomic features of virulence and resistance.

## 2. Materials and Methods

### 2.1. Bacterial Strains

Four hundred fifty *Salmonella* enterica strains representing 49 serotypes were used in this study. All *Salmonella* strains were recovered from diseased livestock and companion animals, including swine (*n* = 229), bovine (*n* = 122), turkey (*n* = 38), chicken (*n* = 30), equine (*n* = 25), caprine (*n* = 4), feline (*n* = 1), and reptile (*n* = 1) sources. They were collected from four state veterinary diagnostic laboratories (AZ, MO, NC, and TN) from 2002 to 2005 (Appendix A). Antimicrobial susceptibility testing (AST) was previously performed using microbroth dilution with 16 antimicrobials (S3). AST profiles of 380 strains recovered from 2002-2003 were previously reported [22]. All isolates were stored at −80 °C in Brucella broth with 20% glycerol until use.

### 2.2. Genome Sequencing, Assembly and Annotation

Genomic DNA was extracted with a QIAamp 96 DNA QIAcube HT kit (Qiagen, Gaithersburg, MD, USA) on the automated high-throughput DNA extraction machine QIAcube HT per the manufacturer’s instructions. The WGS libraries were constructed with Nextera XT kits and then sequenced on the MiSeq platform with v3 reagent kits (Illumina, San Diego, CA, USA) with 2 × 300 bp paired-end reads. The short reads were assembled de novo using CLC Genomics Workbench version 10.0 (Qiagen, Santa Clarita, CA, USA). Among the 448 isolates sequenced on the Illumina platform, the median N50 was 190 kb, ranging from 31 to 718 kb. The median coverage was 53x, ranging from 21× to 131× per genome. Based on antimicrobial resistance, serotype, and source information, forty-three isolates representing 23 serotypes were selected for sequencing on the Pacific Biosciences (PacBio) Sequel platform (PacBio^®^, Menlo Park, CA, USA). Two of the isolates were only sequenced by PacBio and not on the Illumina platform. The continuous-long-reads were assembled by the PacBio Hierarchical Genome Assembly Process (HGAP4.0) program and contigs were circularized by Circlator [28,29]. The size of closed chromosomes ranged from 4,677,885 to 5,033,061 bp. The size of 104 plasmids ranged from 2096 bp to 330,857 bp. The coverage per genome was between 210× and 972×, with an average of 539×. Genomes were annotated using the National Center for Biotechnology Information’s (NCBI) Prokaryotic Genome Automated Pipeline.

### 2.3. Detection of SPIs and Virulence spvRABCD Operon

The bioinformatics pipeline to detect the first 21 SPIs was constructed in-house from a previous study [16]. The additional SPI-22 and 23 sequences were added to the SPI database and bioinformatics pipeline. Briefly, sequences of 23 SPIs were downloaded from GenBank to a local database. Sizes of the SPIs ranged from 1.7 to 133.3 kb, encoding various virulence genes. The existence of SPIs in 450 isolates were identified by performing a BLAST search of the assembled genomes against the SPI local database. The sequence of the *spvRABCD* operon was extracted from the pSDVr (pOU1115) plasmid (Accession DQ115388). A local BLAST search was performed to determine the presence of SPIs and the *spvRABCD* operon for all 450 isolates.

To elucidate the diversity of SPIs, a phylogenetic analysis of SPIs was performed. The distance between different isolates was calculated based on the presence and absence of genes in each SPI and an unweighted pair group method with arithmetic mean (UPGMA) tree was constructed to cluster similar isolates together with PHYLogeny Inference Package (PHYLIP) [26]. The top 9 serotypes comprised 10 or more isolates, and only 10 isolates from each of the top 9 serotypes were selected to generate the SPI phylogenetic tree. Selection of isolates within the same serotypes was based on the source, state, and year of isolation.

### 2.4. Identification of Resistance Genes, Plasmid Types, and SGI

Antimicrobial resistance genes (ARGs), disinfectant resistance genes (DRGs) and heavy metal resistance genes (HMRGs) were identified using the National Center for Biotechnology Information (NCBI) AMRFinderPlus with the default settings (https://github.com/ncbi/amr/wiki/Methods). Plasmid type was determined by PlasmidFinder (https://cge.cbs.dtu.dk/services/PlasmidFinder/) and the sequences were blasted against the database of known plasmid types, selecting for those with at least 90% sequence identity and 60% sequence length. The *Salmonella* genomic island (SGI) is an integrative mobilizable element that is made of a backbone and a multidrug resistance region containing a cluster of genes. The presence of the SGI was detected by blasting genomes against the backbone of SGI based on the SGI structure reported by Hall [30].

### 2.5. Submission and Accession Number

Illumina raw reads from 448 *Salmonella* isolates were submitted to the NCBI (http://www.ncbi.nlm.nih.gov). The accession numbers for SRA are listed in Appendix A. The PacBio raw reads and their assemblies for 43 strains can be accessed through BioSample IDs and accession numbers listed in Appendix A.

## 3. Results

### 3.1. The SPI Profile, Distribution, and Diversity

To elucidate genomic features of virulence, we performed comparative genomic analysis of 450 *Salmonella* strains isolated from diseased animals. SPIs play a crucial role in the interaction between *Salmonella* and host cells, and promote *Salmonella* invasion into epithelial cells. All 446 *Salmonella enterica* subspecies I and two *S. enterica* subspecies IV (*S*. Houtenae) carried the same eight SPIs, including SPI-1 to 6, 9, and 11, and none carried SPI-15, 20, 21, and 22. The presence of other SPIs, including SPI-7, 8, 10, 12, 13, 14, 16, 17, 18, 19, and 23 varied by serotypes (Table 1). For example, SPI-19 was only detected in *S*. Agona, *S*. Enteritidis, *S*. Krefeld, and *S*. Havana. SPI-17 was detected in *S*. Enteritidis, *S*. Krefield and 17% of *S*. Derby. SPI-7 was only detected in 20% of *S*. Muenchen. Most strains within the same serotype shared the same SPI profile; however, some serotypes showed more diverse SPI profiles than others. Among 43 *S*. Newport, only 5% carried SPI-10 and 2% had SPI-18, whereas among 29 *S. Derby* strains, only 17% carried both SPI-8 and SPI-17, meanwhile, 83% carried SPI-23 (Table 1). *S*. Enteritidis carried the most SPIs (*n* = 15), and *S*. Mbandaka, *S*. Cerro, *S*. Meleagridis, and *S*. Havana carried the least (*n* = 10). Two *Salmonella enterica* subspecies IIIa (*S*. Arizona) had different SPI profiles from all other *Salmonella* serotypes, both possessing SPI-1 to 3, 5, 6, 9, 11-14, and 20. SPI-21 was only detected in one *S*. IIIa strain (CVM#22446) from feline. SPI-15 and 22 were not detected in any of the 450 isolates (Table 1).

A phylogenetic tree based on the presence and absence of genes in each SPI showed that, aside from *S*. Choleraesuis, all other serotypes showed great diversity, particularly *S.* Newport, *S.* Derby, *S.* Agona, *S.* Anatum, and *S*. Typhimurium. All *S*. Choleraesuis carried the same genes in each of the 12 SPIs (Appendix A, Figure 1). SPIs were distributed throughout the entire chromosome based on closed genome sequences (Figure 2). Most serotypes showed the distribution of SPIs in similar positions with the same orientation on the chromosome (Figure 2a). However, one *S*. Reading (CVM#35189) was exceptional, as an approximately two million bp DNA segment linked with the SPI-3/SPI-13/SPI-1/SPI-9/SPI-12 sequences had a changed orientation compared with other serotypes (Figure 2b). Further analysis of the break regions of this isolate identified many phage genes.

### 3.2. Resistance Genes

A total of 60 ARGs, 4 DRGs, and 33 HMRGs were identified (Table 2). These ARGs may contribute to resistance to 13 antimicrobial classes, including resistance to aminoglycosides, β-lactams, fosfomycin, macrolides, phenicols, polymyxins, quinolones, sulfonamides, tetracycline, trimethoprim, bleomycin, lincosamides, and streptothricin. Among 450 isolates, 78.4% of isolates (353/450) carried ≥1 ARG, and 69.1% (311/450) carried ≥3 ARGs (Appendix A). Forty-nine (10.9%) isolates from 9 serotypes, including *S*. Agona, *S*. Anatum, *S.* Derby, *S*. Heidelberg, *S*. Newport, *S*. Reading, *S.* Typhimurium, *S*. Uganda, and *S.* Worthington carried ≥10 ARGs. Seven isolates, 2 each of *S*. Agona, *S.* Heidelberg, and *S.* Typhimurium isolated from swine and one *S.* Worthington isolated from bovine carried 14–16 ARGs (Appendix A). Ninety-seven isolates (21.6%), including two *S.* subspecies IIIa and 2 *S.* subspecies IV did not carry any ARGs.

Four DRGs (*qacEdelta1*, *qacE*, *qacL*, *qacG2*) that encoded for resistance to quaternary ammonium compounds were identified, with at least one of these genes found in 44.4% of isolates (200/450) (Appendix A). These DRGs could be located either on plasmids or the chromosome (Appendix A). In addition, 33 HMRGs that encoded resistance to copper, gold, mercurate, tellurite, silver, and arsenic were identified (Appendix A). Two HMRGs, the gold resistance genes *golS* and *golT*, were detected in all 446 *Salmonella enterica* subspecies I isolates, but not in 4 isolates of *Salmonella* subspecies IIIa and IV. Among 446 subspecies I isolates, 50.4% of isolates (225/446) carried additional HMRGs in addition to *golS* and *golT*. These additional HMRGs were mostly located on plasmids, particularly those containing ARGs (Appendix A, Figure 3).

### 3.3. Plasmid Type, Structure and Association with Salmonella Serotypes and Source of Isolates

A total of 12 plasmid types were identified, including IncA/C, IncI, IncN, IncHI, IncFII, IncFIB, IncFIA, IncQ, IncX, IncY, IncL/M, and Col. Eighty-five percent (384/450) of isolates carried at least one plasmid and 65% (293/450) of isolates carried 2 or more plasmid types based on PlasmidFinder (Appendix A). Plasmid type was highly associated with *Salmonella* serotype and source of the isolates. For example, IncA/C was frequently found in *S*. Newport (83.3%) and *S*. Agona (100%) isolated from bovine sources, and IncFII (100%) and IncFIB (100%) were identified in all *S*. Choleraesuis isolated from swine. *S*. Typhimurium isolated from all sources often carried IncFII (88.2%) and IncFIB (86.8%) plasmids. IncX (100%) was detected in all *S*. Kentucky isolated from chicken, and 95% of *S*. Heidelberg isolated from swine carried IncHI. IncQ and IncY were mainly detected in swine isolates (Table 3).

Forty-three isolates, representing 23 serotypes, were sequenced using the PacBio sequel sequencing platform, and chromosomes from 39 isolates were closed. The size of the chromosomes ranged from 4,644,025 to 5,033,061 bp (Appendix A). A total of 104 plasmids were identified among 43 isolates with each isolate carrying 0–9 plasmids. Ninety-four plasmids were circularized. The size of circularized plasmids varied greatly, ranging from 2,096 to 330,859 bp. Twenty-four hybrid plasmids were identified with the most common types being IncHI2-IncHI2A, followed by IncFII-IncFIB. Other hybrid plasmids, including IncFIB-IncHI2A-IncHI2, IncQ-IncHI1B-IncHI1A-IncFIA, IncFIA-IncFII-IncFIB, and IncFIA-IncHI1A-IncHI1B, were also identified (Appendix A). Twenty-nine plasmids could not be assigned to an incompatibility group. IncA/C2 plasmids carried more ARGs than other plasmid types, ranging from 5–12 with an average of 11 ARGs. MDR plasmids often carried multiple HMRGs (Appendix A, Figure 3a,b). One *S*. Typhimurium (O5-) (ID# 28321a) isolated from swine carried 5 plasmids and 3 of them carried either resistance genes or virulence operons (Appendix A). The largest plasmid (p28321a-1) was 219,745 bp in size and is a hybrid plasmid of IncFIA-IncHI1A-InHI1B-IncQ. This plasmid carried 8 ARGs and 2 clusters of mercury resistance genes (*n* = 8). The second plasmid (p28321a-2) was 94,046 bp and typed as IncFII-IncFIB. This plasmid is a virulence plasmid, carrying the virulence operon (*spvRABCD*), but no ARG. The third plasmid (p28321a-3) was 25,682 bp, carried 6 ARGs, 1 DRG, and 7 mercury resistance genes. This plasmid was un-typable based on PlasmidFinder (Figure 3a). The isolate also carried a 42 kb SGI on its chromosome, containing an additional 6 ARGs, including 2 copies of DRGs (*qacEdelta1*) (Figure 3a, Appendix A). One *S*. Heidelberg isolate (ID# 28322) recovered from swine carried 3 plasmids. The largest plasmid (p28322-1) was 234,536 bp, typed as IncHI2-InHI2A, and carried 6 ARGs and 18 HMRGs. The second largest plasmid with size 167,068 bp was identified as an IncA/C plasmid and carried 11 ARGs and 2 copies of DRGs (*qacEdelta1*) plus a cluster of mercury resistance genes (*n* = 7). The smallest plasmid with size 95,358 bp was typed as IncY and had no resistance genes (Figure 3b).

### 3.4. Prevalence of SGI and spv Virulence Operon

SGI is made of a backbone and a class 1 integron that contains a cluster of genes, including multidrug resistance genes. It was present in 4 serotypes, including *S*. Typhimurium, *S*. Derby, *S*. Agona and *S.* Senftenberg with a prevalence of 52%, 52%, 7%, and 38%, respectively (Appendix A). Most *S*. Typhimurium strains with SGI carried the same ARGs, including *aadA2*, *bla*_CARB-2_, *floR*, *sul1*, and *tet(G)* (Appendix A). PacBio sequence data showed that the SGI was located on the chromosome and carried 2 copies of *sul1* and *qacEdelta1* in addition to *floR*, *bla*_CARB-2_, *aadA2*, and *tet(G)* (Appendix A, Figure 3a). One of the *S*. Typhimurium isolates (CVM 24362) carried 33 bp of SGI backbone without any resistance genes based on PacBio sequence data (Appendix A). Instead, this isolate carried 12 ARGs on the 226,966 bp hybrid mega-plasmid, typed as IncQ-IncHI1B-IncHI1A-IncFIA (Appendix A). Several *S*. Derby, *S*. Agona, and *S*. Senftenberg had partial SGI-1 backbones and carried various ARGs (Appendix A). The closed genomes of 2 *S*. Senftenberg (CVM20749 and CVM34514) carried SGIs with 99% sequence homology with 81 kb of *Proteus* genomic island 1 (PGI1) [31]. This resistance island carried 9 resistance genes, including *sul1* (two copies), *dfrA5*, *tet(A)*, *aph(6)-Id*, *aph(3**′′**)-Ib*, *bla*_TEM-1_, *aadA2*, and *ant(2**′′**)-Ia* plus 2 DRGs, *qacEdelta1* and *qacE* (Appendix A).

The virulence operon (*spvRABCD*) was present in 100% of *S*. Choleraeusis (*n* = 35), *S.* I.4,5,12,i:- (*n* = 3) and *S*. Enteritidis (*n* = 4), and 78.7% of *S*. Typhimurium (*n* = 107) (Appendix A). The virulence operon (*spvRABCD*) was often linked with IncFII or hybrid plasmids of IncFII-IncFIB (Appendix A, Figure 3a). Two *Salmonella* IIIa (*S*. IIIa 56:z4 and *S*. IIIa 41:z4) isolates carried unusual *spv* operon with a *spvD* deletion (*spvRABC*) and both *spvRABC* operons were located on the chromosome.

## 4. Discussion

Antimicrobial resistance is one of the greatest public health challenges of our time, resulting in diminishing options for treating bacterial infections in humans and animals. The goal of this study was to elucidate *Salmonella* genomic elements that contribute to virulence and resistance in the context of animal illnesses. We performed WGS and comparative genomic analysis on 450 *Salmonella* strains recovered from diseased animals. The study shed light on distributions of SPIs and the *spvRABCD* locus in different serotypes, and the association between plasmid type, and serotype and source of the isolates. It also provided comprehensive resistance gene profiles and illustrated detailed genomic structure of MDR and plasmids.

*Salmonella* pathogenicity islands contribute to host cell invasion and pathogenesis and can be a key feature contributing to the host specificity of *Salmonella* causing animal infections [8,9]. Suez et al. reported that SPI-1 to 5, 9, 13, and 14 were the part of invasive non-typhoidal *Salmonella* (iNTS) core genome; SPI-6, 10–12, 16–19 were variable among genomes and SPI-7, 8–15 were absent in iNTS [9]. However, our data showed that SPI-6 and 11 were present in all 450 isolates, and at least in our collection were considered as part of the core genome, whereas SPI-13 and 14 were not, because 93 isolates, representing 14 serotypes, did not have SPI-13 and 14. The SPI-7 was detected in 20% of *S*. Muenchen isolates and SPI-8 was detected in 11 different serotypes. We detected all SPIs with exception of SPI-15 and 22 in the current study. It was reported that SPI-22 is unique for *S*. Bongori that has been found predominantly associated with cold-blooded animals, but it can infect humans as well [15]. The differences in SPI profile observed from different studies may be due to methodologies and serotype differences used in these studies. Our work not only has recapitulated findings from early studies but also verified the SPI profiles for certain *Salmonella* serotypes through the use of WGS.

Recently, a study by Monte et al. using WGS data showed that several important *Salmonella* serotypes such as *S.* Typhimurium, *S.* Heidelberg, *S.* Newport, *S.* Infantis, and *S.* Kentucky only carried 3–7 SPIs, with many of them missing SPI-1, and some of them missing SPI-2 or 4 or 5, which are believed to be highly conserved among *Salmonella* serotypes [32]. The significant differences in SPI distribution in this study could be due to differences in the SPI database that were available. Therefore, it is important to have a unified detection methodology, a harmonized database, and a similar bioinformatic pipeline with the same settings.

The SpvRABCD are important virulence factors. Previous studies have shown that *spvRABCD* operon is present in nearly all host-adapted serotypes but only in a few broad-host range serotypes that are highly pathogenic, such as *S*. Typhimurium, and *S*. Enteritidis [33]. The current study showed similar findings. The *spvRABCD* operon was detected in 79% of *S*. Typhimurium, and in all *S*. Choleraesuisis, *S.* I.4.5.12:i:-, and *S.* Enteritidis. The majority of *S*. Typhimurium isolated from swine carried the *spvRABCD* operon which may explain why *S*. Typhimurium is often associated with severe acute enteritis and septicemia in pigs. We also found that two *S*. IIIa carried the *spv* operon with an *spvD* deletion (*spvRABC*) on the chromosome, which is consistent with the previous report [34]. The pathogenicity of this unique virulence operon (*spvRABC*) has not been well studied and needs to be further evaluated.

The *spvRABCD* operon is usually located on a virulence plasmid [17] which has replicons IncX1 and IncFII [35]. Our data showed that an *spv* operon was located on either IncFII or a hybrid plasmid IncFII-InFIB. Previously, we found that, based on Plasmidfinder, *S*. Dublin carried hybrid MDR/virulence plasmids typed as IncA/C2-IncFIA-IncFIB-IncFII-IncX1, IncA/C2-IncFIA, and IncA/C2-IncFII, respectively [16]. AMR-virulence hybrid plasmids have been previously reported in *S*. Typhimurium, *S.* Choleraesuis, and *S*. Enteritidis [35]. Virulence plasmids and AMR plasmid are usually maintained separately in *Salmonella* spp. [36]. It has been reported that virulence plasmids are only transmitted vertically within the same serotype, and rarely transmitted horizontally to other serotypes [37]. A revised food safety risk profile would be necessary if it is found that hybrid MDR-virulence plasmids could transfer horizontally between different serotypes. Further studies are needed to investigate a transferability among hybrid AMR-virulence plasmids.

We have identified 60 ARGs with the potential to encode resistance to thirteen antimicrobial classes. Many of these classes are important to treat *Salmonella* infections in both humans and animals. The traditional antimicrobial agents, such as ampicillin, amoxicillin, and trimethoprim-sulfamethoxazole (TMP-SMZ) were used historically to treat *Salmonella* infections. Due to emerging resistance to these drugs, fluoroquinolones or third-generation cephalosporins are commonly used clinically. The genes that are responsible for resistance to each of these drugs were detected in many of our isolates. For example, >20% of isolates carried the *bla*_CMY-2_ gene that encodes resistance to third-generation cephalosporins, such as ceftriaxone and ceftiofur, two important drugs to treat human and animal salmonellosis, respectively [22]. The *bla*_CMY-2_ gene is often located in MDR IncA/C plasmids that can spread to other serotypes or species by conjugation [38]. Although quinolone resistance is generally low in *Salmonella* in the USA, plasmid-encoded quinolone resistance genes including, *oqxA*, *oqxB*, *qnrB19* were detected in our isolates. Overall, we have identified forty-nine isolates from nine serotypes isolated from bovine, swine, turkey, and equine, which carried ≥10 ARGs. Based on PacBio sequence data, the majority of ARGs were located on plasmids. Mobility of MDR plasmids in different environments is a major public health concern.

We detected a chromosome-encoded SGI in several serotypes that carried multiple ARGs within an SGI cassette. SGI was first reported in *S*. Typhimurium DT104 in 2000 [39]. Since then several other serotypes have been reported to carry SGI [30]. Based on varied genomic structure, different variants of SGI (SGI A to Z) have been described, with a diversity of ARG alleles in MDR regions [30,40]. Our data showed that resistance gene cassettes from the SGI in *S*. Derby, *S*. Agona, and *S*. Senftenberg were different from SGI in *S*. Typhimurium. It is believed that the origin of SGI was from MDR plasmids based on the genomic structure of SGI [30].

In addition to ARGs, DRGs and HMRGs were commonly present among these isolates, and DRGs and HMRGs often co-existed with ARGs in MDR plasmids. A recent report by Yang [41] showed that up to 87% of *Salmonella* isolated from broiler chicken farms and retail meats in China carried HMRGs. Additionally, the HMRGs were significantly associated with the presence of DRGs and ARGs [41]. The significant association between HMRGs with DRGs and ARGs implicated that widely using disinfectants and heavy metals in animal husbandry and food production may play a critical role in co-selecting for antimicrobial resistance development and dissemination.

We observed that plasmid type was highly associated with *Salmonella* serotype and isolate source. This is crucial information for outbreak investigations and foodborne disease source attribution studies. Reports by Folster and others have indicated that although the IncA/C2 plasmid has a broad host range, humans infected with *Salmonella* that carry IncA/C2 linked with *bla*_CMY-2_ were most commonly found among *Salmonella* serotypes usually associated with cattle and beef sources, such as *S*. Newport and *S.* Dublin [27]. Our previous work showed that the IncA/C2 MDR plasmid was commonly present in the bovine-adapted serotype *S*. Dublin [16]. The current study supports such findings. IncA/C was mainly detected in *S.* Newport, *S.* Typhimurium, and *S.* Agona isolated from bovine sources (Table 3). Our data showed that IncA/C plasmids carried more ARGs than other plasmid types with an average of 11 ARGs. A similar finding was reported by McMillan et al., who reported that the IncA/C plasmid subtype ST3 carried ≥4 ARGs, whereas IncX plasmids often carried one or no resistance genes [25]. We also noticed that IncHI, IncQ, and IncY were mainly found in *Salmonella* isolated from swine, and IncFII and IncFIB were commonly found in *S.* Typhimurium and *S.* Choleraesuis.

A total of 12 plasmid types were identified based on PlasmidFinder using MiSeq data. However, when plasmids were closed by PacBio sequencing, 23.5% of them were hybrid plasmids. PacBio sequence data clearly indicated that recombination events have occurred among AMR plasmids and virulence plasmids. Because of the complexity of plasmid recombination, insertion, or deletion events, the use of traditional PCR plasmid typing or using short-read WGS data for plasmid analysis may fail to reveal the full structure of plasmids. With sequencing technology continuing to improve, closing all plasmids has the potential to become reality. Therefore, the development of a new plasmid typing scheme with more detailed subtypes based on this information is possible. Our study shows the advantages of long-read sequencing technology to provide detailed information on the genomic structure of MDR *Salmonella*.

## 5. Conclusions

*Salmonella* isolated from sick animals contained various SPIs and ARGs, virulence plasmids, and MDR plasmids. The plasmid types are associated with the serotype and source of the isolates. Such complicated genomic structures shed light on the strain characteristics contributing to the severity of disease and treatment failures in *Salmonella* infections and has significant human and animal health considerations. WGS offers high resolution for the detection and characterization of full components of virulence and antimicrobial resistance determinants. More studies that include closing plasmids are needed to study MDR and virulence plasmid structure, biology, evolution, and plasmid typing.

## Figures and Tables

**Figure 1 genes-11-01025-f001:**
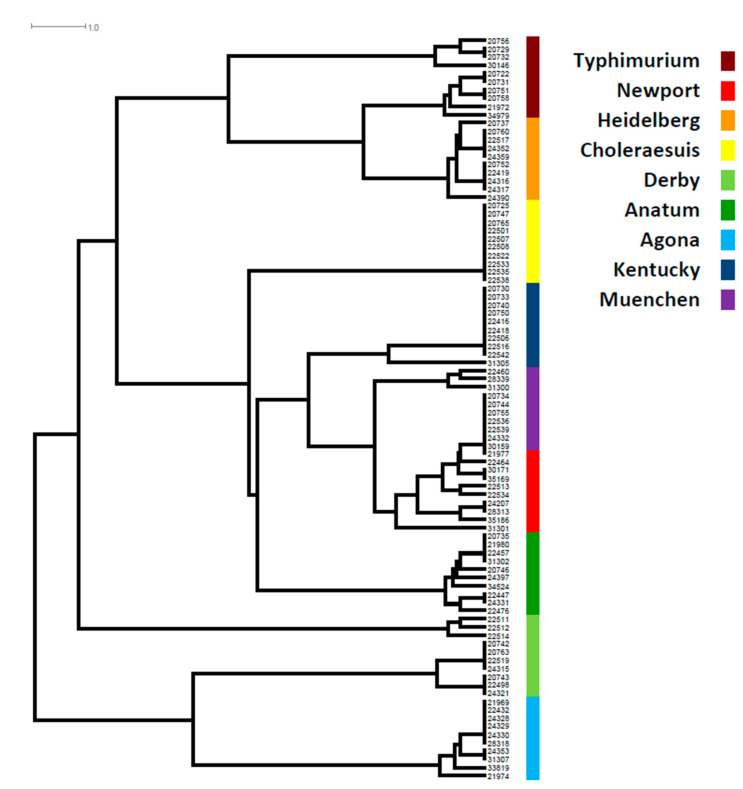
SPI phylogenetic tree in top 9 *Salmonella* serotypes. The color codes are given on the right of the figure. SPI phylogenetic tree was based on presence and absence of genes in each of SPIs and the length of each horizontal line in the tree refers to the number of gene differences among isolates.

**Figure 2 genes-11-01025-f002:**
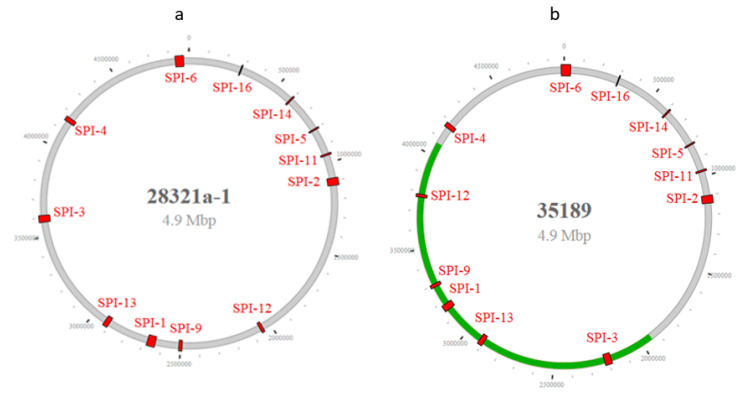
(**a**)**.** Presence and distribution of SPIs in *S*. Typhimurium (28321a) and (**b**). *S*. Reading (35189) genomes. Green line indicated that a DNA segment linked with the SPI-3/SPI-13/SPI-1/SPI9/SPI-12 sequences had a changed orientation compared with other serotypes.

**Figure 3 genes-11-01025-f003:**
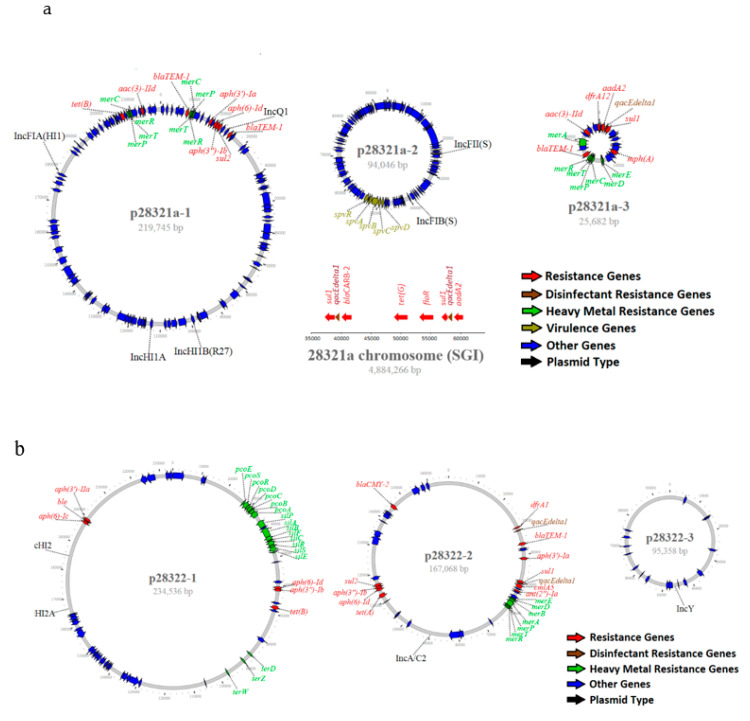
(**a**) Genomic structure of MDR plasmids and virulence plasmids from *S.* Typhimurium (O5-) (28321a) and (**b**). *S*. Heidelberg (28322) isolated from diseased swine. The color codes are given on the bottom right of each figure.

**Table 1 genes-11-01025-t001:** Presence of *Salmonella* pathogenicity islands (SPI) in different *Salmonella* serotypes ^1^.

Serotype	SPI 1	SPI 2	SPI 3	SPI 4	SPI 5	SPI 6	SPI 7	SPI 8	SPI 9	SPI 10	SPI 11	SPI 12	SPI 13	SPI 14	SPI 15	SPI 16	SPI 17	SPI 18	SPI 19	SPI 20	SPI 21	SPI 22	SPI 23
**Typhimurium (136) ^2^**	**+**	**+**	**+**	**+**	**+**	**+**	**-**	**-**	**+**	**-**	**+**	**+**	**+**	**+**	**-**	**+**	**-**	**-**	**-**	**-**	**-**	**-**	**-**
**Newport (43)**	**+**	**+**	**+**	**+**	**+**	**+**	**-**	**-**	**+**	**5%**	**+**	**+**	**+**	**+**	**-**	**+**	**-**	**2%**	**-**	**-**	**-**	**-**	**-**
**Heidelberg (38)**	**+**	**+**	**+**	**+**	**+**	**+**	**-**	**-**	**+**	**-**	**+**	**+**	**+**	**+**	**-**	**+**	**-**	**-**	**-**	**-**	**-**	**-**	**-**
**Choleraesuis (35)**	**+**	**+**	**+**	**+**	**+**	**+**	**-**	**-**	**+**	**-**	**+**	**+**	**+**	**+**	**-**	**+**	**-**	**-**	**-**	**-**	**-**	**-**	**-**
**Derby (29)**	**+**	**+**	**+**	**+**	**+**	**+**	**-**	**17%**	**+**	**-**	**+**	**+**	**-**	**-**	**-**	**+**	**17%**	**-**	**-**	**-**	**-**	**-**	**83%**
**Anatum (16)**	**+**	**+**	**+**	**+**	**+**	**+**	**-**	**-**	**+**	**87%**	**+**	**+**	**+**	**+**	**-**	**-**	**-**	**-**	**-**	**-**	**-**	**-**	**+**
**Agona (14)**	**+**	**+**	**+**	**+**	**+**	**+**	**-**	**+**	**+**	**-**	**+**	**+**	**-**	**-**	**-**	**+**	**-**	**-**	**+**	**-**	**-**	**-**	**+**
**Kentucky (13)**	**+**	**+**	**+**	**+**	**+**	**+**	**-**	**+**	**+**	**-**	**+**	**+**	**-**	**-**	**-**	**+**	**-**	**-**	**-**	**-**	**-**	**-**	**-**
**Muenchen (11)**	**+**	**+**	**+**	**+**	**+**	**+**	**20%**	**-**	**+**	**-**	**+**	**+**	**+**	**+**	**-**	**+**	**-**	**-**	**-**	**-**	**-**	**-**	**27%**
**Worthington (9)**	**+**	**+**	**+**	**+**	**+**	**+**	**-**	**+**	**+**	**-**	**+**	**+**	**-**	**-**	**-**	**+**	**-**	**-**	**-**	**-**	**-**	**-**	**-**
**Mbandaka (9)**	**+**	**+**	**+**	**+**	**+**	**+**	**-**	**-**	**+**	**-**	**+**	**+**	**-**	**-**	**-**	**+**	**-**	**-**	**-**	**-**	**-**	**-**	**-**
**Uganda (9)**	**+**	**+**	**+**	**+**	**+**	**+**	**-**	**-**	**+**	**-**	**+**	**+**	**+**	**+**	**-**	**-**	**-**	**-**	**-**	**-**	**-**	**-**	**-**
**Senftenberg (8)**	**+**	**+**	**+**	**+**	**+**	**+**	**-**	**+**	**+**	**-**	**+**	**+**	**-**	**-**	**-**	**+**	**-**	**-**	**-**	**-**	**-**	**-**	**-**
**Oranienburg (7)**	**+**	**+**	**+**	**+**	**+**	**+**	**-**	**-**	**+**	**-**	**+**	**-**	**+**	**+**	**-**	**+**	**-**	**+**	**-**	**-**	**-**	**-**	**86%**
**I 4,5,12: i: -(3)**	**+**	**+**	**+**	**+**	**+**	**+**	**-**	**-**	**+**	**-**	**+**	**+**	**+**	**+**	**-**	**+**	**-**	**-**	**-**	**-**	**-**	**-**	**-**
**Schwarzengrund (6)**	**+**	**+**	**+**	**+**	**+**	**+**	**-**	**-**	**+**	**-**	**+**	**-**	**+**	**+**	**-**	**+**	**-**	**+**	**-**	**-**	**-**	**-**	**-**
**Muenster (5)**	**+**	**+**	**+**	**+**	**+**	**+**	**-**	**-**	**+**	**+**	**+**	**-**	**+**	**+**	**-**	**-**	**-**	**+**	**-**	**-**	**-**	**-**	**+**
**Reading (5)**	**+**	**+**	**+**	**+**	**+**	**+**	**-**	**-**	**+**	**-**	**+**	**80%**	**+**	**+**	**-**	**+**	**-**	**20%**	**-**	**-**	**-**	**-**	**-**
**Montevideo (4) ^3^**	**+**	**+**	**+**	**+**	**+**	**+**	**-**	**-**	**+**	**-**	**+**	**-**	**+**	**+**	**-**	**+**	**-**	**+**	**-**	**-**	**-**	**-**	**-**
**Enteritidis (4)**	**+**	**+**	**+**	**+**	**+**	**+**	**-**	**-**	**+**	**+**	**+**	**+**	**+**	**+**	**-**	**+**	**+**	**-**	**+**	**-**	**-**	**-**	**-**
**Saintpaul (4)**	**+**	**+**	**+**	**+**	**+**	**+**	**-**	**-**	**+**	**-**	**+**	**+**	**+**	**+**	**-**	**+**	**-**	**-**	**-**	**-**	**-**	**-**	**-**
**Give (4)**	**+**	**+**	**+**	**+**	**+**	**+**	**-**	**-**	**+**	**-**	**+**	**-**	**+**	**+**	**-**	**-**	**-**	**+**	**-**	**-**	**-**	**-**	**-**
**Bredeney (3)**	**+**	**+**	**+**	**+**	**+**	**+**	**-**	**-**	**+**	**-**	**+**	**-**	**+**	**+**	**-**	**+**	**-**	**+**	**-**	**-**	**-**	**-**	**-**
**Hadar (3)**	**+**	**+**	**+**	**+**	**+**	**+**	**-**	**-**	**+**	**-**	**+**	**+**	**+**	**+**	**-**	**+**	**-**	**-**	**-**	**-**	**-**	**-**	**-**
**Albany (2)**	**+**	**+**	**+**	**+**	**+**	**+**	**-**	**+**	**+**	**-**	**+**	**+**	**-**	**-**	**-**	**+**	**-**	**-**	**-**	**-**	**-**	**-**	**-**
**Braenderup (2)**	**+**	**+**	**+**	**+**	**+**	**+**	**-**	**-**	**+**	**-**	**+**	**+**	**+**	**+**	**-**	**+**	**-**	**-**	**-**	**-**	**-**	**-**	**-**
**Cerro (2)**	**+**	**+**	**+**	**+**	**+**	**+**	**-**	**-**	**+**	**-**	**+**	**+**	**-**	**-**	**-**	**+**	**-**	**-**	**-**	**-**	**-**	**-**	**-**
**Hartford (2)**	**+**	**+**	**+**	**+**	**+**	**+**	**-**	**-**	**+**	**-**	**+**	**+**	**+**	**+**	**-**	**+**	**-**	**-**	**-**	**-**	**-**	**-**	**-**
**Infantis (2)**	**+**	**+**	**+**	**+**	**+**	**+**	**-**	**-**	**+**	**-**	**+**	**+**	**+**	**+**	**-**	**+**	**-**	**-**	**-**	**-**	**-**	**-**	**-**
**Java (2)**	**+**	**+**	**+**	**+**	**+**	**+**	**-**	**-**	**+**	**-**	**+**	**+**	**+**	**+**	**-**	**+**	**-**	**-**	**-**	**-**	**-**	**-**	**-**
**Tennessee (2)**	**+**	**+**	**+**	**+**	**+**	**+**	**-**	**+**	**+**	**-**	**+**	**+**	**-**	**-**	**-**	**+**	**-**	**-**	**-**	**-**	**-**	**-**	**-**
**Bovismorbificans (1)**	**+**	**+**	**+**	**+**	**+**	**+**	**-**	**-**	**+**	**-**	**+**	**+**	**+**	**+**	**-**	**+**	**-**	**-**	**-**	**-**	**-**	**-**	**-**
**Cubana (1)**	**+**	**+**	**+**	**+**	**+**	**+**	**-**	**+**	**+**	**-**	**+**	**+**	**-**	**-**	**-**	**+**	**-**	**-**	**-**	**-**	**-**	**-**	**-**
**Gaminara (1)**	**+**	**+**	**+**	**+**	**+**	**+**	**-**	**-**	**+**	**-**	**+**	**-**	**+**	**+**	**-**	**-**	**-**	**+**	**-**	**-**	**-**	**-**	**-**
**Javiana (1)**	**+**	**+**	**+**	**+**	**+**	**+**	**-**	**-**	**+**	**-**	**+**	**-**	**+**	**+**	**-**	**+**	**-**	**+**	**-**	**-**	**-**	**-**	**-**
**Johannesburg (1)**	**+**	**+**	**+**	**+**	**+**	**+**	**-**	**-**	**+**	**-**	**+**	**-**	**+**	**+**	**-**	**-**	**-**	**+**	**-**	**-**	**-**	**-**	**-**
**Krefeld (1)**	**+**	**+**	**+**	**+**	**+**	**+**	**-**	**+**	**+**	**-**	**+**	**-**	**-**	**-**	**-**	**+**	**+**	**-**	**+**	**-**	**-**	**-**	**-**
**Litchfield (1)**	**+**	**+**	**+**	**+**	**+**	**+**	**-**	**-**	**+**	**+**	**+**	**+**	**+**	**+**	**-**	**+**	**-**	**-**	**-**	**-**	**-**	**-**	**-**
**Lille (1)**	**+**	**+**	**+**	**+**	**+**	**+**	**-**	**+**	**+**	**-**	**+**	**+**	**-**	**-**	**-**	**+**	**-**	**-**	**-**	**-**	**-**	**-**	**-**
**London (1)**	**+**	**+**	**+**	**+**	**+**	**+**	**-**	**-**	**+**	**-**	**+**	**+**	**+**	**+**	**-**	**-**	**-**	**-**	**-**	**-**	**-**	**-**	**-**
**Meleagridis (1)**	**+**	**+**	**+**	**+**	**+**	**+**	**-**	**+**	**+**	**-**	**+**	**+**	**-**	**-**	**-**	**-**	**-**	**-**	**-**	**-**	**-**	**-**	**-**
**Miami (1)**	**+**	**+**	**+**	**+**	**+**	**+**	**-**	**-**	**+**	**-**	**+**	**-**	**+**	**+**	**-**	**+**	**-**	**+**	**-**	**-**	**-**	**-**	**-**
**Norwich (1)**	**+**	**+**	**+**	**+**	**+**	**+**	**-**	**-**	**+**	**-**	**+**	**+**	**+**	**+**	**-**	**+**	**-**	**-**	**-**	**-**	**-**	**-**	**-**
**Panama (1)**	**+**	**+**	**+**	**+**	**+**	**+**	**-**	**-**	**+**	**-**	**+**	**-**	**+**	**+**	**-**	**+**	**-**	**+**	**-**	**-**	**-**	**-**	**-**
**Poona (1)**	**+**	**+**	**+**	**+**	**+**	**+**	**-**	**-**	**+**	**+**	**+**	**-**	**+**	**+**	**-**	**+**	**-**	**+**	**-**	**-**	**-**	**-**	**+**
**Havana (1)**	**+**	**+**	**+**	**+**	**+**	**+**	**-**	**-**	**+**	**-**	**+**	**-**	**-**	**-**	**-**	**+**	**-**	**-**	**+**	**-**	**-**	**-**	**-**
**Arizona (IIIa 56:z4) (1)**	**+**	**+**	**+**	**-**	**+**	**+**	**-**	**-**	**+**	**-**	**+**	**+**	**+**	**+**	**-**	**-**	**-**	**-**	**-**	**+**	**+**	**-**	**-**
**Arizona(IIIa 41:z4) (1)**	**+**	**+**	**+**	**-**	**+**	**+**	**-**	**-**	**+**	**-**	**+**	**+**	**+**	**+**	**-**	**-**	**-**	**-**	**-**	**+**	**-**	**-**	**-**
**Houtenae (IV 43:z4) (2)**	**+**	**+**	**+**	**+**	**+**	**+**	**-**	**-**	**+**	**-**	**+**	**+**	**+**	**+**	**-**	**-**	**-**	**-**	**-**	**-**	**-**	**-**	**-**

^1^ Green: SPIs presented in all *S*. *enterica* subspecies I and IV isolates; Yellow: SPIs are absent in all *S*. *enterica* subspecies I and IV isolates; no color: SPIs presence varied by serotypes; ^2^ Typhimurium included *S.* Typhimurium, *S.* Typhimurium (O5-) and variant of *S.* Typhimurium.

**Table 2 genes-11-01025-t002:** Identified antimicrobial, biocide, and heavy metal resistance genes from 450 *Salmonella* strains.

Resistance Class	Genes
Aminoglycoside	*aadA1*, *aadA2*, *aadA3*, *aadA5*, *aadA6*, *aadA7*, *aadA12*, *aadA13*, *aadA15*, *aadA22*, *aadA25*, *aph(3′)-Ia*, *aph(3′)-IIa*, *aph(3′′)-Ib*, *aph(4)-Ia*, *aph(6)-Id*, *aph(6)-Ic*, *ant(2′′)-Ia*, *aac(3)-VIa*, *aac(3)-IId*, *aac(3)-IV*, *aac(3)-IIa*, *aac(6′)-Ib*, *aac(6′)-Ib4*
β-lactam	*bla*_CARB-2_, *bla_CMY_*, *bla_CMY-2_*, *bla_HER-3_*, *bla_TEM_*_-1_, *bla*_TEM_, *bla*_OXA-9_
Fosfomycin	*Fos*, *fosA7*
Macrolide	*erm(B)*, *mph(A)*
Polymyxins	*mcr-9.1*
Phenicol	*catA1*, *cmlA1*, *cmlA5*, *cmlA6*, *floR*
Quinolone	*oqxA*, *oqxB*, *qnrB19*
Sulfonamide	*sul1*, *sul2*, *sul3*
Tetracycline	*tet(A)*, *tet(B)*, *tet(C)*, *tet(G)*, *tet(M)*
Trimethoprim	*dfrA1*, *dfrA5*, *dfrA12*, *dfrA14*
Bleomycin	*ble*, *bleO*
Lincosamide	*lnu(G)*
Streptothricn	*sat2*
Biocide resistance	*qacEdelta1*, *qacE*, *qacL*, *qacG2*
Heavy Metal Resistance	*golS*, *golT*, *merA*, *merB*, *merC*, *merD*, *merE*, *merF*, *merP*, *merT*, *merR*, *pcoA*, *pcoB*, *pcoC*, *pcoD*, *pcoE*, *pcoR*, *pcoS*, *silA*, *silB*, *silC*, *silE*, *silF*, *silP*, *silR*, *silS*, *terD*, *terE*, *terW*, *terZ*, *arsA*, *arsB-mob*, *arsD*

**Table 3 genes-11-01025-t003:** Plasmid type associated with *Salmonella* serotypes and source of isolates.

Serotype ^1^	Source	Plasmid Type										
		IncA/C	IncI	Col	IncN	IncHI	IncFII	IncFIB	IncFIA	IncQ	IncX	IncY
Typhimurium ^2^ (*n* = 136)	Bovine (32)	6	6	1	0	0	25	22	1	0	0	0
	Swine (84)	1	11	40	0	5	75	76	16	3	3	2
	Chicken (5)	0	2	2	0	0	5	5	0	0	1	0
	Turkey (5)	1	3	2	0	0	5	5	1	0	0	0
	Equine (9)	0	0	0	0	0	9	9	0	0	0	0
	Caprine (1)	0	1	0	0	0	1	1	0	0	0	0
	Total	8	23	45	0	5	120	118	18	3	4	0
Newport (*n* = 43)	Bovine (30)	25	5	4	1	0	5	2	0	0	2	0
	Swine (5)	1	0	0	0	0	0	0	0	0	0	0
	Equine (7)	0	0	0	0	0	2	1	0	0	0	0
	Caprine (1)	0	0	0	0	0	0	0	0	0	0	0
	Total	26	5	4	1	0	7	3	0	0	2	0
Heidelberg (*n* = 38)	Swine (20)	0	10	3	0	19	0	0	0	0	0	3
	Turkey (13)	2	7	0	1	7	0	0	0	0	3	0
	Chicken (5)	0	2	4	0	0	0	0	0	0	5	0
	Total	2	19	7	1	27	0	0	0	0	8	0
Choleraesuis (*n* = 35)	Swine (35)	0	0	1	0	25	35	35	0	33	1	0
Derby (*n* = 29)	Swine (29)	0	13	14	0	3	1	1	4	6	5	4
Anatum (*n* = 16)	Swine (9)	0	6	9	1	1	0	0	0	0	0	0
	Bovine (5)	2	1	1	0	0	1	1	1	0	0	0
	Equine (1)	0	0	0	0	1	0	0	1	1	0	0
	Caprine (1)	0	1	0	0	0	0	0	0	0	0	0
	Total	2	8	10	1	2	1	1	2	1	0	0
Agona (*n* = 14)	Bovine (5)	5	3	1	0	0	0	0	0	0	0	0
	Swine (8)	6	1	2	1	0	0	0	0	0	0	0
	Turkey (1)	0	1	0	0	0	0	0	0	0	0	0
	Total	11	5	3	1	0	0	0	0	0	0	0
Kentucky ^3^ (*n* = 13)	Bovine (2)	0	1	0	0	0	0	0	0	0	0	0
	Chicken (10)	0	6	0	0	3	6	5	0	0	10	0
	Swine (1)	0	0	0	0	0	0	0	0	0	0	0
	Total	0	7	0	0	3	6	5	0	0	10	0
Muenchen (*n* = 11)	Bovine (2)	0	0	0	0	0	0	1	0	0	0	0
	Swine (9)	0	1	6	0	0	0	0	0	0	0	0
	Total	0	1	6	0	0	0	1	0	0	0	0

^1^ Only included the top serotypes that comprised 10 or more isolates, ^2^ Typhimurium included *S.* Typhimurium, *S.* Typhimurium (O5-), and variants of *S.* Typhimurium. ^3^ One *S*. Kentucky isolate recovered from chicken (30172) carried IncL/M.

## Data Availability

WGS data for all isolates listed in this study are available from NCBI. The accession numbers to access these data are listed in Appendix A.

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
