# Peer review of "Comparative Genomic Analysis of 450 Strains of Salmonella enterica Isolated from Diseased Animals"

_genes, 2020, doi:10.3390/genes11091025_

Round 1

Reviewer 1 Report

Zhao et al. report on the SPI profile, resistance genes, and plasmids (type, structure and association with serotype) in 450 Salmonella enterica strains. The samples were sequenced on a MiSeq using a 2x300 library, and 43 isolates were also sequenced using PacBio long-read sequencing. Both the "Introduction" and the "Discussion" section are well written. However, the results seem to be presented without a discuss of their impact, which should be improved.   Suggested major changes:   1. The title of the manuscript should reflect that only S. enterica strains were analyzed.   2. Section 2.5: It is not clear why the authors mention that only 448 of the 450 sequences were submitted to NCBI.   3. Section 3.1: The authors report an inversion in a S. Reading sample. Could they comment on whether the breakpoint was resolved, and if there are genomic features overlapping those breakpoints?   4. Section 3.3: The authors report that 12 plasmid types were identified using the MiSeq information. Table 3 contains 10 plasmids, and I am not sure why 2 of them are not included in the table.   5. Section 3.3: The authors report that 12 plasmid types were identified using the sequences from MiSeq. When they sequenced 43 of those isolates using PacBio, they identified 104 plasmids. In the discussion, the authors mention the drawbacks of using short-read sequencing for plasmid profiling, but it is still not clear to me why these numbers vary so much between the technologies. Were these plasmids not captured in the Illumina sequences, or were the authors unable to detect them due to methodological limitations (for example, due to the sequence identity thresholds used)?    6. The term Salmonella Genomic Island (SGI) should be defined before use in the manuscript.   7. In the discussion, the authors speculate that the significant differences they observe in SPI distribution compared to Monte et al. 2019 could be due to differences in the SPI database that were available. Did the authors verify that this was the case? I would suggest running their pipeline on the Monte et al. data (or a subset of the data) to determine whether or not this was the case.    7. In the discussion, the authors mention that several of the plasmids closed using PacBio sequencing were hybrids. This is an interesting finding and should be included in the abstract.   Suggested minor changes:   1. Table 1 can be included in the supplement and/or summarised as a heatmap.   2. Section 3.4: The authors mention "Instead, this isolate carried 12 ARGs on the 226,966 pb mega-hybrid plasmid, typed as IncQ-IncHI1B-IncHI1A-IncFIA (S2).". The "226,966 pb" should be changed to "226,966 bp".   3. Supplementary Table S2: The name of the sheet is misspelt and should be changed to "Summary".  

Reviewer 2 Report

The manuscript titled “Comparative Genomic Analysis of Four Hundred Fifty Salmonella Strains Isolated from Diseased Animals” describes an investigation of 450 Salmonella strains with focus on pathogenicity islands and resistance genes. Of the 450 isolates, 43 were (randomly?) selected for PacBio sequencing, yielding complete or near-complete genomes. Most of the bioinformatic analyses are quite standard and simple, although a custom database for SPIs were created. It would be nice to make this custom database available for others e.g. as a supplemental fasta file.

The 450 Salmonella isolates were recovered from 2002 and 2005 and may thus not represent the newest “snapshot” of Salmonella genomes with regards to ARGs and SPIs. This is a drawback in the study. Otherwise, the results are thorough enough and the study presents fairly relevant insight into Salmonella resistance genes with a few interesting observations.

My biggest issue with the manuscript is the lack of coupling the resistance gene predictions with the results of the antimicrobial susceptibility testing, that is mentioned in the “Methods and Methods” section (which should obviously be corrected). When you have made the MIC tests and have predicted ARGs, it is a shame not to couple these results. It is very often the case that identified ARGs do not actually confer resistance or lead to high MIC values, but you have the opportunity to discuss this aspect here.

Generally, the manuscript would benefit from more polishing. The Introduction and Discussion sections are quite well written and well structured, but Methods and Results sections need to be rewritten in multiple places, in order to clarify the conveyed messages.

Figure 3 is truncated and it is not explained why there is a part A and a part B.

Other comments:

Page 2: The meaning of the following phrase is not really clear

”Several serotypes such as S. Dublin, S. Choleraesuis, S. Agona, S. Uganda showed 100% resistance to >1 antimicrobial.”

Does it mean that 100% of the tested serotypes were resistant? Or is it related to a very high MIC value. Please rephrase and consider to mention how many antibiotics were tested.

Page 3: How and/or why were the 43 isolates chosen for PacBio?

Page 3: The software ”Circlator” is misspelled as ”Circulator”

Page 3: Please specify what kind of data is meant with the 210x-972x coverage.

Some sections of the methods are not clearly rewritten and should be revised. This includes the last paragraph of section 2.3 and most of 2.4. E.g. ”Presence of SGI was detected by blasting genomes against with backbone of SGI genes”

Page 7: The entire paragraph in the top should be revised for grammar. What is the meaning of this sentence: ”SPIs distributed throughout the entire chromosome based on closed genomes (Figure 2).”?

Figure 1: Please indicate what the branch lengths are.

Figure 2: It is not obvious from the text (page 7) that figure 2a is an example of one of the serotypes from ”Most of serotypes showed the distribution of SPIs in the similar position with the same orientation on chromosome (Figure 2a).”

Figure 2: Figure text should also be revised for grammar.

Figure 3: The figure is truncated in the manuscript and it is not known how much is missing.

There is no description of what Figure 3a and Figure 3b shows.

In the figure legend, ”Resistance Genes” should be changed to ”Antibiotic Resistance Genes”.

”Plasmid Type” is not a good enough descriptor. Please change.

Why and how were the examples in figure 3 chosen?

In the first sentence of section 3.4, it sounds like there is only one SGI version. Please revise to reflect that there are many types of SGIs.

Page 12: The sentence “PacBio sequence data showed that SGI was located on the chromosome and carried two copies of sul1 and qacEdelta1 in additional to floR, blaCARB-2, aadA2 and tet(G) (S2, Figure 3a).” is misleading. This result is not from PacBio sequence data, but observed in the assembly that is based in PacBio data (and Illumina).

Page 12: This sentence “One of the S. Typhimurium isolates (CVM 24362) carried 33 bp of SGI backbone without any resistance genes based on PacBio sequence data (S2)” should also be revised as above. Also, 33 bp is very short – it is more likely kbp? Please also indicate that this is the chromosome. At least, I assume it is, since you go on to describe a megaplasmid in the next sentence.

Page 12: Change “226,966 pb” to “226,966 bp”

Page 12: In the sentence: “The closed genome of two S. Senftenberg (CVM20749 and CVM34514) carried a SGI that had 99% sequence homology with 81 kb of proteus genomic island 1 (PGI1)” it sounds like there is a partial match (81 kbp) to PGI1, but PGI1 is 81.1 kbp in total, so it is a full length hit. Also, please change kb to kbp.

Page 14: Please rephrase the sentence “A recent report by Yang (Yang et al. 2020) showed that Salmonella isolated from broiler chicken farms and retail meats in China carried up to 87% HMRGs.” To make it reflect that it was 87% of Salmonella isolates has HMRGs.

Page 14: The Conclusion section is very generic here and does not summarise any of the interesting findings. This should be improved so that the most interesting finding(s) are mentioned. Sentences like “WGS offers high resolution for detection and characterization of full components of virulence and antimicrobial resistance determinants. More studies that include closing plasmids are needed to study MDR and virulence plasmid structure, biology, evolution and plasmid typing.” are redundant – especially in Conclusion – in 2020.

Author Response

Please see the attachment below

Reviewer 3 Report

The submitted manuscript presents the results of a study involving the sequencing of 450 genomes of Salmonella. The outcomes are relevant in the context of studying animal illnesses. The manuscript is well-written and easy to follow. In my opinion, there are only minor points that need to be addressed before the publication:  

Page 2: There is a typo in “it has become possible to comprehensively identify SPIs and other genomic futures” (features).

Page 4: Please revise the sentence “Presence of SGI was detected by blasting genomes against with backbone of SGI genes (Hall 2010)”.

Table 1: The explanation of indices “1” (in the title, next to “serovars”) and “2” (next to “Typhimurium (136)”) is not included in the footnote.

Figure 2: I suggest including the names “28321a-1” and “35189” in the figure caption (they are depicted in the plasmids, but not referred to in the figure caption). This comment also applies to figure 3.

Table 2: Please make sure that the names of the genes are always separated by commas (the comma is missing in some of the rows). Moreover, the gene names “lnu(G)” and “sat2” are not written in italics.

Figure 3: Whatever the reason may be, figure 3 is not displayed correctly in the PDF version (a part of the figure is missing). Please pay attention to this issue.

Page 11: I suggest changing “The size of a chromosome ranged from 4,644,025-5,033,061 bp.” to “The size of a chromosome ranged from 4,644,025 to 5,033,061 bp.”

Page 12: I suggest using “Proteus genomic islands” (starting with a capital letter).

Page 13: A space missing in “historicallyto”.

The quantity is sometimes given in words, sometimes with the use of numerals, e.g. “Forty-three isolates, representing 23 serotypes…”(page 11). I suggest using the numerals throughout the text (“43 isolates, 23 serotypes…”). This also applies to the title of the manuscript (“Comparative Genomic Analysis of 450 Salmonella Strains Isolated from Diseased Animals”).

Author Response

Please see the attachment below.

Round 2

Reviewer 1 Report

The authors have addressed all of my major concerns. There are a few minor errors, e.g., "Illumina" is misspelt as "illumine" in section 2.2, which can easily be fixed.

Reviewer 2 Report

Many of my concerns have been addressed or have been welll argued against. However, some other comments were ignored or only briefly addressed. 

The English language in "Methods and methods" and results section have not been improved and are, in a few places, a hindrance to proper understanding of the text. This raises a concern that the native English speaking authors did not carefully read and edit the manuscript. Of course, the language does not have to be perfect, but there are many very obvious spelling errors and grammatical mistakes, especially in methods and results (e.g. the header "Methods and Methods", "Nextara", "illumine" and so on).

I am still missing the MIC or similar values for the tested strains, which could easily be included as supplementary information. 
